# THE BABYVIEW DATASET: HIGH-RESOLUTION EGOCENTRIC VIDEOS OF INFANTS' AND YOUNG CHILDREN'S EVERYDAY EXPERIENCES

## ABSTRACT

Human children far exceed modern machine learning algorithms in their sample efficiency, achieving high performance in key domains with much less data than current models. This "data gap" is a key challenge both for building intelligent artificial systems and for understanding human development. Egocentric video capturing children's experience – their "training data" – is a key ingredient for comparison of humans and models and for the development of algorithmic innovations to bridge this gap. Yet there are few such datasets available, and extant data are low-resolution, have limited metadata, and importantly, represent only a small set of children's experiences. Here, we provide the first release of a large developmental egocentric video dataset – the BabyView dataset – recorded using a high-resolution camera with a large vertical field-of-view and gyroscope/accelerometer data. This 430 hour dataset includes egocentric videos from children spanning 6 months – 5 years of age in longitudinal, at-home contexts. We provide gold-standard annotations for the evaluation of speech transcription, speaker diarization, and human pose estimation, and evaluate models in each of these domains. We train self-supervised language and vision models and evaluate their transfer to out-of-distribution tasks including syntactic structure learning, object recognition, depth estimation, and image segmentation. Although performance in each scales with dataset size, overall performance is relatively lower than when models are trained on curated datasets, especially in the visual domain. Our dataset stands as an open challenge for robust, human-like AI systems: how can such systems achieve human-levels of success on the same scale and distribution of training data as humans?

## 1 INTRODUCTION

Infants and young children are remarkable learners, becoming capable and engaged social partners within their first two years of life. The pace of this developmental progress far exceeds modern machine learning algorithms in its efficiency and capacity (Frank, 2023). In particular, signature accomplishments of artificial systems such as few-shot learning (Brown et al., 2020) and image classification (Krizhevsky et al., 2012) require hundreds of billions of words of training data and millions of labeled images. In contrast, human learners become proficient in extending labels for newly learned visual concepts (Carey & Bartlett, 1978) and producing language (Frank et al., 2021) from only tens of millions of words and far fewer labeled examples (Zhuang et al., 2021). This "data gap" between human and machine learners is thus a key challenge for the joint goals of understanding human learning and building intelligent artificial systems. Making progress will require not just an understanding of the flexibility of human intelligence, but also an understanding of the efficiency of human learning.

Data availability is a major barrier to progress in our understanding of the gap in learning efficiency between machines and humans. To make effective comparisons between human and machine learners, we need to be able to evaluate models on data comparable to what children see and hear during everyday learning experiences. While models are trained on millions of images and/or videos, these are taken from the adult perspective, providing a very different vantage point on the world that is disconnected from real-world learning environments.

Egocentric video recordings taken from the child's perspective provide a key window into what children both see and hear as they learn about the world around them and from their social partners (Smith et al., 2015; Yoshida & Smith, 2008; Aslin, 2009; Franchak et al., 2011). Developmental psychology studies using these types of video recordings have together revealed that the infant view is dramatically different from that of an adult (Yoshida & Smith, 2008) and varies as children learn to locomote on their own and interact actively with the objects, places, and people around them (Kretch et al., 2014; Long et al., 2022).

Here we present the largest high-resolution developmental egocentric video dataset to date, the BabyView dataset. We collect videos from 28 families predominantly from around the U.S., totalling 430 hours of usable recordings. We capitalize on innovations in the development of head-mounted cameras (Long et al., 2023), obtaining videos with a large vertical field of view and coordinated gyroscope/accelerometer data that can be used to estimate the child's own head movements. We provide pose detection, automated speech transcriptions, and diarization, along with gold-standard annotations for use in evaluating each of these. We then evaluate self-supervised vision and language models on these data relative to existing benchmarks.

## 2 RELATED WORK

**Few developmental egocentric video datasets are available** Egocentric video has been an important domain for computer vision (Damen et al., 2022; Grauman et al., 2022) and resulting commercial applications, such as wearable devices. Yet egocentric video datasets are mostly taken from the adult perspective, including the Ego4D dataset, which has become an important standard in this field (Grauman et al., 2022). Head-mounted cameras have also been used in research with children, including both descriptive investigations (Yoshida & Smith, 2008; Aslin, 2009; Franchak et al., 2011; Kretch et al., 2014; Fausey et al., 2016; Bergelson & Aslin, 2017) and computer vision studies (Sheybani et al., 2024; Zhuang et al., 2021). Unfortunately, most prior work did not obtain consent for broad sharing with other research groups and so many major datasets are unavailable for re-analysis.

Those developmental egocentric video datasets that are available have been difficult to use for training models for reasons of both data quantity and quality (Long et al., 2022; Sullivan et al., 2021; Bergelson & Aslin, 2017). For example, the SAYCam dataset – by far the largest available dataset – is relatively low-resolution (480 x 640 pixels), has limited motion-correction (leading to blurry views) and has timestamps imprinted on every frame (Sullivan et al., 2021). The audio quality is quite variable depending on the background noise and context, and the videos have restricted vertical view angle that obscures views of children's hands and what children are interacting with. Further, SAYCam represents video from three children of highly-involved and informed academic parents, all of whom were the first children in their families. These issues have limited the field's ability to make use of automated annotations of the visual or linguistic content of these videos and have restricted the ability to use these data to draw broadly generalizable conclusions. Here we present the largest high-resolution, developmental egocentric video dataset with broad consent from caregivers for reuse within the research community.

**Models trained on developmental data show limited performance** Self-supervised vision models trained using developmental egocentric video data (Zhuang et al., 2021; Orhan et al., 2020; Zhuang et al., 2022; Orhan & Lake, 2024; Vong et al., 2024) have had some intermediate success. However, these representations trained from egocentric videos significantly underperform those self-supervised models trained on curated datasets, while the latter models approach the accuracy of models trained using fully-supervised methods (Oquab et al., 2023; Caron et al., 2021; He et al., 2021; Chen et al., 2020; He et al., 2020). Thus, it remains unclear whether the current state-of-the-art techniques represent truly general purpose visual learning algorithms. In particular, it is unclear whether gaps in model performance are due to dataset quality and quantity or instead due to the difficulty of learning robust representations from children's more realistic everyday inputs.

Relatedly, in the language domain, recent work has investigated the possibility of training language models (LMs) on small-scale developmental datasets (see e.g., Warstadt et al., 2023; Zhuang et al., 2024; Feng et al., 2024), but most of these have focused on datasets larger than those available from egocentric video data. For example, the text data used in the popular BabyLM competition (Warstadt et al., 2023) are also meant to approximate what a 10-year-old child could receive (including text

Table 1: The BabyView dataset is the only egocentric developmental video dataset with accelerometer/gyroscope data that is available for research.

| Dataset | Ego? | Long? | Type | N | Hours | Audio | Transcript | Motion |
|---|---|---|---|---|---|---|---|---|
| BV-Home | ✓ | ✓ | Infant | 28 | 433 | ✓ | ✓ | ✓ |
| Ego-SingleChild | ✓ | ✓ | Infant | 1 | 47 | ✓ | ✓ | |
| SAYCam Sullivan et al. (2021) | ✓ | ✓ | Infant | 3 | 476 | ✓ | ✓ | |
| Ego4D Grauman et al. (2022) | ✓ | | Adult | 931 | 3,670 | ✓ | ✓ | |
| Epic Kitchens Damen et al. (2018) | ✓ | | Adult | 37 | 100 | ✓ | ✓ | |

from Wikipedia and other sources), which is very likely more – and different – data than what is required to acquire a language. One exception is Qin et al. (2024), who trained GPT-2 (Radford et al., 2019) on very small amounts of input from a single child and investigated the amount of grammatical knowledge that could be learned.

Here, we evaluate whether data from a new, high-resolution dataset will lead to increases in performance for self-supervised visual and linguistic benchmark models.

## 3 THE BABYVIEW DATASET

We address gaps in data availability by collecting and analyzing a new set of developmental egocentric videos: the BabyView dataset. The current paper describes the first release of the dataset, but data collection is still ongoing and we anticipate future growth in the overall size of the dataset. Recordings were obtained using a high-resolution head-mounted camera for infants and children from 6 months through 5 years of age in both at-home and preschool settings. In the BabyView-Home portion of the dataset, 28 families recorded longitudinal data during everyday activities for a total of 433 hours across all children. All videos are accompanied by accelerometer/gyroscope data that can be used to estimate children's head-motion (Joshi et al., 2010; Karpenko et al., 2011; Joshi et al., 2022). We additionally release the Ego-SingleChild dataset, a related dataset with a different camera (see below). Together, these data comprise the first release of the largest high-resolution egocentric video dataset from the child perspective that will be available to researchers for both descriptive analysis and model building (see Table 1 for comparison to prior datasets).

### 3.1 CAMERA AND SENSOR DATA

The BabyView camera is a GoPro Hero Bones camera attached to a child-safety helmet. This camera was selected because it has gyroscope and accelerometer data, built-in image stabilization features, and relatively high resolution sound and video (Long et al., 2023). The camera is oriented vertically and is neutral with respect to the face plane of the child, enabling the camera to capture both adult faces and objects within a child's hands in the same image, with an effective view angle of 100° vertical by 75° horizontal (see Figure 1a,b)) (Long et al., 2023).

### 3.2 DATASET COMPONENTS

**BV-Home** Twenty-eight families consented to capture home recordings with their infant-toddler (0;5-3;1 years, average age at onboarding = 11 months, SD = .50 years, see Figure 1c). Families were recruited from a convenience sample of researchers in the field of cognitive development (N=9/28 families) and from local advertisements within the State of California. Some English-speaking and English/Spanish bilingual families (N=16/28) completed parent-report measures of children's language development using the long-forms of the MacArthur-Bates Communicative Development Inventories (Marchman et al., 2023; Jackson-Maldonado et al., 2003). See SI for further information on participant consent, detailed demographics, and language questionnaires.

**Ego-SingleChild** We also release 47 hours of data from a single child of an academic who recorded frequently. They used a Cigno F18 Night Vision 1080P Headband Sport Camera rather than the BabyView camera, which yields shorter and lower-resolution videos.

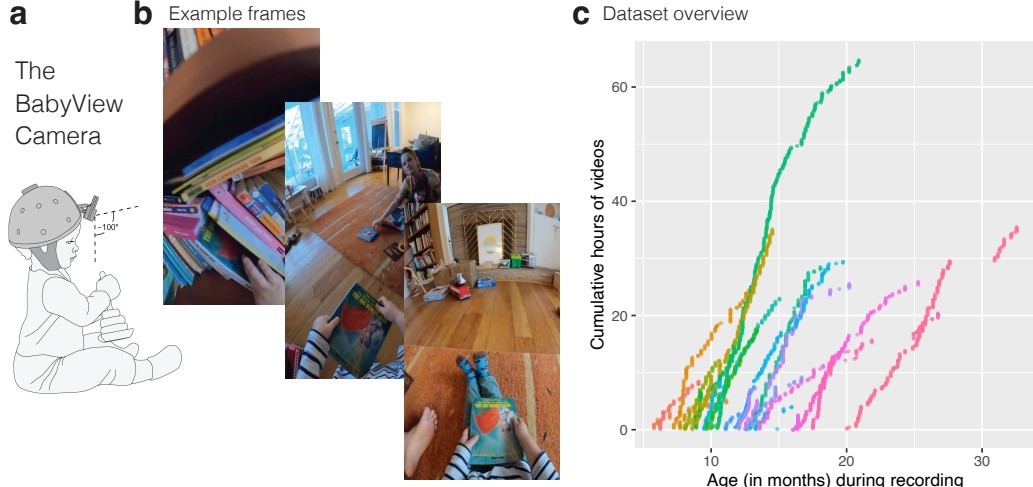

Figure 1: (a) Schematic of a child wearing the BabyView camera illustrating a large vertical field of view. (b) Example frames from a video in the dataset. (c) Cumulative hours of video by each of the participants in the BV-Home subset of the dataset; each color represents an individual child. Data collection is ongoing.

### 3.3 DATA ACCESS & ONGOING DATA COLLECTION

Egocentric video data from children in their home and school environments necessarily contain more sensitive information than videos in egocentric videos by adults. Families provide full consent for the data that are shared at the time of recording and also have a 6 month period after recording when they can retract any portion of their recording. Thus, all data in this release will be made available in November 2024 once the parental embargo period has lapsed. To ensure BabyView data are accessible to researchers while protecting the privacy of participants, we distribute the data through Databrary (https://nyu.databrary.org/) (Gilmore et al., 2016), similar to previous developmental egocentric datasets (Sullivan et al., 2021; Bergelson & Aslin, 2017). Databrary is an US National Institutes of Health-funded site designed specifically for the distribution of developmental video data. Access to data on Databrary requires investigators be authorized via an institutional agreement that bars reidentification of participants and redistribution of data.

BabyView is an ongoing longitudinal project and our aim is to release further data as the dataset grows. Because of the multi-faceted and growing nature of our dataset, we do not pre-specify train/test splits, recognizing that any split might be appropriate for only a subset of research goals (e.g., examining age-related change, or within- vs. cross-child change).

## 4 ANNOTATIONS

### 4.1 LANGUAGE ANNOTATIONS

**Transcription & diarization pipeline**   All videos were transcribed using Distil-Whisper-large-v3.[1] As this version only supports English transcription, we discarded utterances for transcription validation that were in languages other than English (BV-Home, $N$=643 utterances, 24.82%). We also ran a multilingual voice type classifier (Lavechin et al., 2020) on the audio extracted from all BabyView-Home videos, which classified the speech segments as originating from a female adult, male adult, key child (the wearer of the camera), or other child. Each utterance was assigned to one speaker by choosing the model-annotated speaker category that had the greatest overlap with the utterance timestamps. In some cases, an utterance did not overlap with any model-annotated speaker; these were marked as NA (NA rate was 7.18% for BV-Home). For our language model training experiments below, we also ran the same pipeline on the SAYCam audio, though we did not conduct validation on this dataset.

---

[1]Available at https://huggingface.co/distil-whisper.

Table 2: Language annotation results across the age of the child and the speaker. Child-produced speech and infant-directed speech had the highest error rates.

| Dataset | Child age | Speaker | WER | Diarization precision | Diarization recall | $N$ |
|---|---|---|---|---|---|---|
| BV-Home | All Ages | All Speakers | 0.38 | 0.61 | 0.61 | 1947 |
| | 6-18 m.o. | Adult | 0.30 | 0.79 | 0.66 | 1103 |
| | | Key-child | 1.11 | 0.48 | 0.72 | 190 |
| | | Other-child | 0.51 | 0.39 | 0.64 | 88 |
| | 18-30 m.o. | Adult | 0.37 | 0.77 | 0.64 | 271 |
| | | Key-child | 0.56 | 0.62 | 0.76 | 94 |
| | | Other-child | 0.21 | 0.38 | 0.60 | 15 |

**Evaluation procedure** We hand-annotated a subset of 1947 utterances, stratified across age and participant. Two authors transcribed the speech and labeled the speaker in each segment ($N$=1.61 hours). For transcription validation, we computed a Word Error Rate (WER), which is is the ratio of the number of word-level errors to the total number of words in the original utterance Gandhi et al. (2023). To evaluate speaker diarization accuracy, we computed precision and recall of the model output by age and speaker.

**Child-produced and child-directed speech is challenging for transcription algorithms** WER for automated transcriptions was comparable to typical adult performance in the preschool classroom recordings (see Sparks et al. (2024)), but somewhat lower in the naturalistic home environments. Qualitatively, these decrements in performance appear to result from a high prevalence of infant-directed speech that annotation algorithms are less familiar with. Although automated transcriptions perform poorly for the youngest children, we see considerable improvement in WER of child-produced speech of toddler children. The speaker diarization algorithm (Lavechin et al., 2020) was able to identify whether a child vs. adult was speaking 77% of the time, and often could accurately identify the speaker type in the accompanying audio (see Table 2). While combining speaker diarization and automated transcriptions can be very useful, modern transcription algorithms are still considerably less accurate than humans at understanding both child-directed and child-produced speech.

## 4.2 HUMAN POSE ANNOTATIONS

**Pose annotations** We evaluated how well state-of-the-art pose detectors perform on the BabyView dataset. To do so, we first sampled 353 frames from the dataset (stratified across participants and sessions) and manually annotated the 333 non-blurry frames using LabelStudio (Tkachenko et al., 2020-2022), creating a validation set. To efficiently annotate the frames, we deployed the RTMPose (Jiang et al., 2023) model via MMPose (Contributors, 2020a) as a backend to provide initial pose keypoints and bounding box predictions, which we then manually corrected. The pose annotations followed the format used in the COCO keypoints dataset (Lin et al., 2014; Sun et al., 2019). To evaluate the accuracy of keypoint detections and compare our results with those of other studies, we adopted the Object Keypoint Similarity (OKS) metric, as used by (Sun et al., 2019) (details in SI).

**Child egocentric viewpoints are challenging for most pose detection models** The BabyView validation set was more challenging for most models than the COCO validation set (Lin et al., 2014), highlighting a new pose benchmark for naturalistic egocentric videos (see Table 3). However, ViTPose-H, the largest model in the group, showed comparable performance between the two validation sets, suggesting that it is more robust to viewpoint variation.

Table 3: Pose Detection performance on COCO2017 Val and BabyView Val. BabyView Validation frames were more challenging the COCO for all models except ViTPose-H.

| Architecture | #Params | Input Size | COCO AP | BV AP | COCO AR | BV AR |
|---|---|---|---|---|---|---|
| RTMO-l (Lu et al., 2023) | 44.8M | 640x640 | 0.724 | 0.593 | 0.762 | 0.723 |
| YOLOXPose-l (Maji et al., 2022) | 87.0M | 640x640 | 0.712 | 0.588 | 0.749 | 0.658 |
| SIMCC-resnet50 (Li et al., 2022) | 25.7M | 384x288 | 0.735 | 0.676 | 0.790 | 0.723 |
| RTMPose-l-aic-coco (Jiang et al., 2023) | 36.7M | 384x288 | 0.773 | 0.735 | 0.819 | 0.773 |
| HRFormer-pose-base (YUAN et al., 2021) | 43.2M | 384x288 | 0.774 | 0.743 | 0.823 | 0.785 |
| ViTPose-H (Xu et al., 2022) | 632M | 256x192 | 0.788 | 0.788 | 0.840 | 0.825 |

## 5 BENCHMARKS

### 5.1 LANGUAGE REPRESENTATION LEARNING

Next, inspired by the BabyLM challenge, which seeks to learn human-like linguistic representations from small amounts of developmentally-realistic data (Warstadt et al., 2023), we examined the ability to learn linguistic representations from the BV-Home transcripts. For contrast, we compare with high-quality data from the Child Language Data Exchange System (CHILDES), a repository of human-transcribed corpora of children and caregivers' talk (MacWhinney, 2014).

**Experiment Setup** We pretrained GPT-2 (Radford et al., 2019) with 124M parameters (small) on each dataset for up to 20 epochs (see SI for details). After deduplication, the automatically-transcribed utterances for BV-Home and SAYCam each consisted of ∼2M total words. For contrast, the total amount of human-transcribed English-language data available in CHILDES is ∼20M words. Hence, we sampled 2M words of conversation from CHILDES (2.4M total words including speaker labels and other metadata) to align the amount of training data across datasets. We then separated each dataset into train and validation splits, using an 85/15 split. We further compared with training on the combination of BV-Home and SAYCam data and ∼4M words of conversation (4.8M total words) from CHILDES. We also trained a version on the entirety of the English subset of CHILDES (∼20M words), in line with Feng et al. (2024). For evaluation, we used Zorro (Huebner et al., 2021), a benchmark compatible with child vocabulary that aims to quantify the grammatical knowledge of LMs by assessing their capability to effectively distinguish between minimal pairs of sentences that exhibit various grammatical contrasts.

**BV-Home transcriptions provide comparable learning signal for grammatical knowledge** All GPT-2 models achieved above-chance performance on the Zorro evaluation, even with only ∼2M words of training data (see SI for complete results). With 2M words, there was only a negligible difference between BV-Home (64.13%) and SAYCam data (64.06%) and a minor advantage for CHILDES (66.57%). However, combining BV-Home and SAYCam led to matched performance (69.39%) to CHILDES 4M (69.76%). Training on the full CHILDES English subset of 20M words resulted in significantly higher performance (77.77%), as expected with much more language data. This is also shown in Figure 2; training on more language data results in better performance, in contrast to our vision data scaling experiments shown in Figure 3. Overall, despite the potential data quality issues in BabyView and SAYCam transcripts (introduced by multilingual data and speech recognition errors), we observe that transcriptions of BV-Home and SAYCam are comparable to CHILDES as a learning signal for language models to obtain grammatical knowledge.

### 5.2 VISUAL REPRESENTATION LEARNING

We conducted a first set of experiments to investigate the ability of recent self-supervised models to learn useful visual representations from frames taken from these egocentric videos. Enabled by BV-Home, we conduct the largest scale evaluation to date of self-supervised learning methods trained on children's egocentric visual experience.

**Experiment Setup** We trained a ViT-B/14 DINOv2 (Oquab et al., 2023) from scratch as our reference self-supervised learning algorithm, due to its high performance on a variety of downstream tasks, including object recognition, depth estimation and semantic segmentation. We used the standard

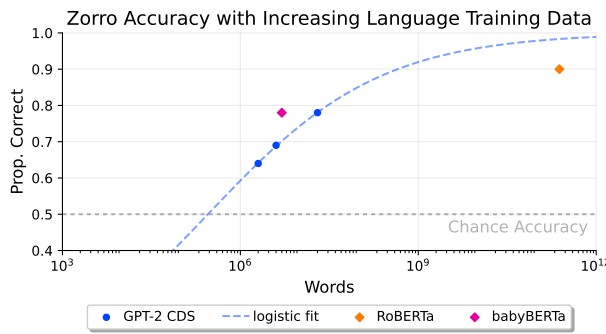

Figure 2: Language data scaling experiments, showing grammatical accuracy on the Zorro benchmark (chance = 0.5) for GPT-2 trained on progressively increasing amounts of child-directed speech (CDS) language data. Within the GPT-2 CDS data points, the first represents 2M words from the BV-Home corpus, the second represents 4M words combined over the BV-Home and SAYCam corpora, and the final point represents 20M words from the CHILDES corpus. Zorro accuracy is also shown for RoBERTa (Liu et al., 2019) [240M words] and BabyBERTa (Huebner et al., 2021) [5M words].

Table 4: Object recognition, depth estimation, and semantic segmentation results on the BabyView & comparison datasets. Downstream generalization accuracy is significantly reduced when learning on frames from egocentric videos relative to curated datasets.

| Dataset | Object Recognition – Top 1 | | Depth Estimation | Semantic Segmentation |
| | ImageNet kNN | ImageNet linear | NYUv2 RMSE↓ | COCOStuff mIoU↑ |
| --- | --- | --- | --- | --- |
| None (random init.) | 10.00 | 1.43 | 0.886 | 0.54 |
| LVD-124M (Oquab et al., 2023) | 82.10 | 84.50 | 0.307 | 44.46 |
| ImageNet (Russakovsky et al., 2015) | 76.29 | 77.64 | 0.456 | 34.65 |
| Ego4D(Grauman et al., 2022) | 43.59 | 54.39 | 0.525 | 23.78 |
| SAYCam(Sullivan et al., 2021) | 42.59 | 52.52 | 0.518 | 21.08 |
| BV-Home | 40.72 | 52.19 | 0.526 | 22.03 |
| SAYCam + BV-Home | 41.76 | 53.28 | 0.511 | 22.53 |

training configuration from the official code base across all training runs. We sampled Ego4D at 1 FPS, leading to 15M frames, and sampled the BV-Home and SAYCam at 5FPS, leading to about 8M frames per dataset. Despite the inherent redundancy in video data, this ensured a relatively large amount of data, compared with the 1.4M ImageNet training set. We evaluated object recognition accuracy on ImageNet, and after additional training on high-resolution images of the original datasets, we evaluate depth estimation on NYUv2 (Silberman et al., 2012) and semantic segmentation on COCOStuff (Caesar et al., 2018). On top of the frozen ViT, for ImageNet we use kNN and a linear probe, whereas for depth estimation we trained a DPT and for semantic segmention we used a linear probe, following the DINOv2 protocols.

**Self-supervised learning from any egocentric data is challenging** We anticipated that the more diverse and higher-resolution videos in BV-Home would afford improvements over prior egocentric video datasets (Sullivan et al., 2021). Yet we found that models trained on BV-Home data did not outperform those trained on the SAYCam dataset, despite the difference in data quality (see Table 4), though we found a small improvement in semantic segmentation performance on models trained on BV-Home vs. SAYCam.[2] More broadly, however, we found that the gap in performance is not just specific to data collected from children. Even when training on Ego4D – a roughly 7x larger and more diverse dataset – we see that a significant gap to curated vision datasets remains across all tasks. We further investigated training an additional self-supervised learning method, MoCov3 (Chen et al., 2021) also based on a ViT-B/16 on the full dataset. We obtained 18.7 for kNN and 27.3 for linear on ImageNet, indicating that other self-supervised learning techniques also show a significant gap in performance.

---

[2]Note results are above random chance: ImageNet – 0.001, NYUv2 – 2, COCOstuff – 0.2.

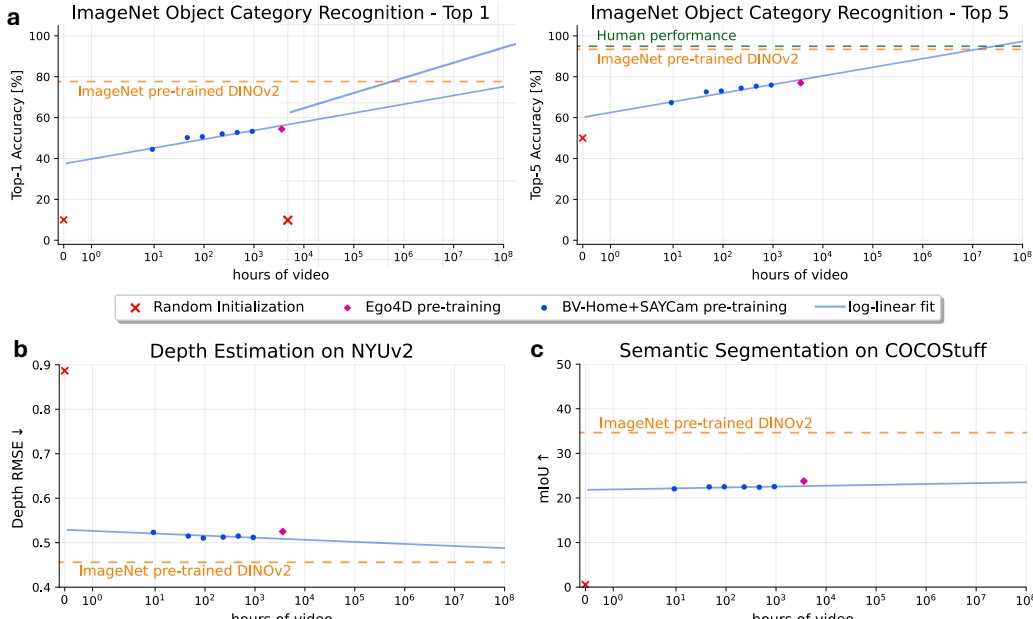

Figure 3: Data scaling experiments for object recognition, depth estimation and semantic segmentation. In **a** we observe a trend that DINOv2 would require upwards of $10^7$ hours of video to match human or ImageNet self-supervised ImageNet performance. In **b** and **c** we also observe unfavorable scaling for depth estimation and semantic segmentation.

**Insufficient scaling to meet human or self-supervised performance from curated datasets**
Given a reasonably large amount of training data from egocentric video of children's visual experience, could the current self-supervised state-of-the-art obtain equivalent performance to training on curated vision datasets or human performance? We trained on 1%, 5%, 10% 25%, 50% and 100% of a combined dataset of BV-Home and SAYCam, and extrapolate by fitting log-linear trend lines. For object recognition on ImageNet (see Figure 3a) we observed that more than $10^7$ hours would be required to reach human performance (Russakovsky et al., 2015) or ImageNet pre-training performance. In Figures 3b and 3c, we find that a similar trend holds for depth estimation and semantic segmentation, with saturating performance as the scale of data is increased. Note that the first two points on these plots indicate 160K and 800K images, and the last point 16M images. While a similar "data gap" finding has also been reported by Orhan (2021), our new dataset and models yield a somewhat lower estimation of the amount of data needed to achieve human-level performance.

## 6 GENERAL DISCUSSION

We present a new, large-scale high-resolution egocentric video dataset documenting infants' and young children's everyday experiences, accompanied by both dense metadata and gold-standard annotations for several key domains. In contrast to prior work with lower-resolution videos and earlier models (Long et al., 2022), we find that state-of-the-art speech recognition (Gandhi et al., 2023; Radford et al., 2023) and pose detection (Xu et al., 2022; Contributors, 2020a) models perform well on stratified samples of frames and audio recordings from the dataset. Further, language models trained on these data performed comparably to models trained on current gold-standard corpora of hand-transcribed speech. The new BabyView camera thus provides improved data over which supervised algorithms can extract descriptives that will be an important resource for characterizing children's linguistic and social learning environments (Sparks et al., 2024).

Yet our results also suggest that the naturalistic, everyday experiences of children pose a challenging problem for the most advanced of our learning algorithms, especially in the visual domain: current state-of-the-art models fall short relative to existing benchmarks when trained on "human amounts" of visual or linguistic data, requiring unrealistic amounts of additional data to achieve human-level performance (Frank, 2023). In particular, our results suggest that current self-supervised visual

learning models are dependent on large, curated datasets with a broad diversity of inputs to construct robust representations.

What might lead to more child-like models of early learning? One idea is that the joint learning of visual and language representations requires more fine-grained and efficient learning algorithms, such as lexicon-level visual grounding (Zhuang et al., 2023; 2024). Further, children's everyday experience contains deep regularities within activity contexts (Clerkin et al., 2017; Clerkin & Smith, 2022; de Barbaro & Fausey, 2022) that are challenging for current models but appear advantageous for human learners. Constructing models that can learn as children do from these skewed input distributions is thus a key challenge for future work. We further speculate that focusing on modeling event-representations in naturalistic video (Zhuang et al., 2020), children's own head-motion via IMU data (Joshi et al., 2022), and attentional guidance from caregivers (Long et al., 2022; Yu et al., 2021) may yield more data-efficient models of early learning.

Our results highlight the need for developmentally appropriate outcome data with which we can be used to evaluate models trained on developmental data. Toddlers cannot classify all ImageNet categories, and a growing literature suggests that object recognition abilities mature throughout middle childhood (Long et al., 2024; Huber et al., 2023). Systematically comparing models' and children's emerging representations may help elucidate the observed gap in model performance.

These data have several limitations. First, these data necessarily incorporate selection bias: parents who opt-in to the study are recording in their homes when they choose to (to avoid privacy issues) and can choose to excise any portion of their data; some naturalistic experiences (e.g., bathtime) are not incorporated into the dataset. Further, with two exceptions, all families are located in the United States, limiting generalizability. Nonetheless, BV-Home incorporates data from a greater diversity of families across race, ethnicity, and family incomes than before (see SI). The potential harms that could arise from this dataset relate to breaches of privacy and trust on the part of the participating families. To guard against these, researchers are required to sign the Databrary data use agreement (Gilmore et al., 2016), which prohibits reidentification or redistribution of videos.

In sum, we present the first release of a new, large-scale, high-resolution developmental egocentric video dataset. Our dataset stands as a challenge to modern AI: how can how can such systems achieve human levels of success on the same scale and distribution of training data as human children?

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

AUTHOR CONTRIBUTIONS

BLINDED.

ACKNOWLEDGEMENTS

We gratefully acknowledge the participating families without whom this work would not be possible. This work was funded by [BLINDED]. We thank many research assistants who have played a key role in the construction of this dataset, including [BLINDED].

# A  APPENDIX

## A.1  DATASET DETAILS

### A.1.1  PARTICIPANT CONSENT

All data collection was approved under [BLINDED] and consent was obtained via one-on-one conversations. Given the sensitive nature of the data, families had multiple opportunities to withdraw their recordings. They could mark videos for deletion during recording and up to six months during the embargo period.

### A.1.2  PARTICIPANT INSTRUCTIONS & RECORDING DETAILS

All participant instructions were taken from Long et al. (2023) which developed the protocols for using the BabyView Camera, and are publicly available at https://osf.io/kwvxu/.

Families were instructed to record as often as was feasible for their families, with a requested minimum of 45 minutes per week. We use standard, rechargeable 9V battery to provide power to the BabyView camera, which allows for continuous 45-60 minute recordings on a standard charge. Families were then compensated based on the duration (mins) of video recordings they provided on a weekly basis as well as bonuses for questionnaires, totalling 18,370.00 dollars across all families.

### A.1.3  BV-HOME ADDITIONAL PARTICIPANT DEMOGRAPHICS

Our sample is highly educated, with 21/28 families having at least one parent with a graduate degree, and with all families having at least one parent with a 4-year college degree. 11/28 children are exposed to more than one language at home, including the following languages: English, Chinese, Farsi, French, Gujarati, Japanese, Korean, Malayalam, Portuguese, Spanish, Tagalog, Thai, Vietnamese. Geographically, 20/28 of families live within California, 4/28 live in the Northeastern United States, 1/28 live in the Southern United States, 1/28 live in the Midwestern United States, 1/28 live in Canada, and 1/28 live in South Korea.

Participating children were 64.29% female, 35.71% male, 0.0% African American/Black, 17.86% Asian American/Pacific Islander, 42.89% Caucasian/White, 10.71% Hispanic/Latinx, 39.29% multiracial, 0.0% other.

We only have income information for 25/28 families, as reporting was optional. The average family income of our sample is 221,143 USD (75,000–1,000,000 USD, SD = 201,710 USD). 13/25 families have more than one child in the household, 1/25 families live in a single-parent household, and 2/25 families have more than 2 caregivers living in the household.

### A.1.4  BV-HOME LANGUAGE OUTCOME QUESTIONNAIRES

Long-form MacArthur Bates CDI language questionnaires (https://mb-cdi.stanford.edu/) were administered every 3 months starting at enrollment. Families were provided compensation for each questionnaire. These parent-report forms assess children's language comprehension and production; aggregate data by age can be viewed at wordbank.stanford.edu. Forms were administered through Web-CDI (https://webcdi.org/). A total of 28 (2 Spanish, 26 English) questionnaires are included in this first release of the dataset.

### A.1.5 VIDEO PROCESSING PIPELINE

Videos were manually uploaded by each family to their personalized Google Drive folders. The uploaded videos were automatically downloaded to a secure server where the metadata (accelerometer and gyroscope) were extracted and the videos were compressed then uploaded to a second Google Drive platform. The compression step used the ffmpeg (Tomar, 2006) program to encode video into the libx265 format with a constant rate factor of 23 to enable high quality MP4 videos.

## A.2 ANNOTATION DETAILS

### A.2.1 POSE KEYPOINT DETAILS AND EVALUATION

The pose keypoints that were evaluated includes 17 keypoints: nose, left eye, right eye, left ear, right ear, left shoulder, right shoulder, left elbow, right elbow, left wrist, right wrist, left hip, right hip, left knee, right knee, left ankle, and right ankle.

The Object Keypoint Similarity (OKS) metric reported is as follows:

$$OKS = \frac{\sum_i \exp\left(-\frac{d_i^2}{2s^2 k_i^2}\right) \delta(v_i > 0)}{\sum_i \delta(v_i > 0)}.$$

In this formula, $d_i$ represents the Euclidean distance between the detected keypoint and the ground truth, $v_i$ indicates the visibility of the ground truth keypoint, $s$ denotes the object scale, and $k_i$ is a constant specific to each keypoint that adjusts the falloff. We report standard metrics for average precision and recall: AP (the average of AP scores at 10 different OKS thresholds: 0.50, 0.55, ..., 0.90, 0.95), and AR (the average of AR scores at OKS = 0.50, 0.55, ..., 0.90, 0.95).

### A.2.2 COMPUTE RESOURCES AND INFRASTRUCTURE FOR ANNOTATIONS

Our annotation work was performed on an internal cluster server with an AMD EPYC 9334 32-Core Processor, 756GB memory, 8 NVIDIA A40 GPUs, and Ubuntu 20.04. We used 8 GPUs for speech recognition and 1 GPU for both assisting with annotation and testing pose detection models on the validation set.

## A.3 LANGUAGE BENCHMARK DETAILS

### A.3.1 LANGUAGE MODEL TRAINING & EVALUATION DETAILS AND DATA PROCESSING

In training our GPT-2 models, we used a learning rate (LR) of 1e-04, linear LR scheduler with no warmup steps, a batch size of 16 per GPU, seed of 42, and Adam optimizer with $\beta = (0.9, 0.999)$ and $\epsilon = 1e - 08$.

The final chosen GPT-2 model for each dataset is the epoch that performed best (had the lowest loss) on the corresponding validation split. The corresponding tokenizer for each model was also trained from scratch on the corresponding dataset.

The training data was set up so that each line corresponded to a single transcribed conversation, which is broken up into chunks of 1024 consecutive tokens by GPT-2 during training. To ensure the data format is consistent for evaluation purposes, we aligned the most important and frequently occurring speaker labels across datasets (mainly based on the existing CHILDES labels): CHI for the target child, MOT for the mother or female adult, and OCHI for other children. All other speaker labels were kept to their default. Around 60% or more of all utterances within each dataset were from CHI or MOT.

See below for an example of part of a single training conversation. Double asterisks surround speaker labels, double newline tokens separate utterances, and an end-of-text token marks the end of the conversation. This format was consistent across all conversations and datasets.

*\*\*CHI\*\*: Hi. \n\n \*\*CHI\*\*: There you go. \n\n \*\*OCHI\*\*: Do you have a little ball in your cup. \n\n (...) \n\n \*\*CHI\*\*: Are those your stars? \n\n \*\*MOT\*\*: Can you say star? \n\n*

*\*\*CHI\*\*: Star. \n \n \*\*CHI\*\*: Look. \n \n \*\*CHI\*\*: Stars. \n \n \*\*MOT\*\*: Stars. See? Look, look at the yellow star, a golden star. <|endoftext|>*

We found cases of duplicate conversations and duplicate utterances within conversations among the transcribed data across the three datasets. We removed these to the best of our ability before training.

The Zorro evaluation was inspired by BLiMP (Warstadt et al., 2020) and is a modification for child-directed language (e.g. lower vocabulary). However, it was designed specifically for masked language models such as RoBERTa. To adapt it to GPT-2, we reformatted the Zorro data to match the BLiMP format and used the BLiMP evaluation in the BabyLM evaluation suite [3] since the main difference between the two is the evaluation data. Further, we use the full Zorro test suite and do not filter examples by vocabulary. Hence, our results are not comparable to Qin et al. (2024) which filters Zorro examples by the vocabulary of their training datasets.

To better match the training data format and assess the effects of speaker labels on evaluation, we came up with three variations of Zorro: 1) the original Zorro evaluation sentences, 2) the sentences with the CHI speaker label prepended, and 3) the sentences with the MOT speaker label prepended. To further match the training data, the speaker labels were surrounded by double asterisks, and sentences included double newline tokens (before and after).

As seen in Table 5, all models perform better when the evaluation data is more closely aligned with the training data format (2nd or 3rd variation of Zorro sentences), especially with the MOT speaker label (3rd variation). This is likely because the utterances spoken by the mother or female adults are typically more grammatical than those of the child.

### A.3.2 DETAILED LANGUAGE MODEL EXPERIMENT RESULTS

See Table 5 for the Zorro evaluation results of our GPT-2 models, along with the best Zorro evaluation format for each.

Table 5: Quantitative results on the Zorro benchmark

| Model | Zorro (Final Avg.) | Best Evaluation Format |
|---|---|---|
| BV-Home | 64.13% | CHI |
| SAYCam | 64.06% | MOT |
| CHILDES (2M) | 66.57% | MOT |
| SAYCam + BV-Home | 69.39% | CHI |
| CHILDES (4M) | 69.76% | MOT |
| CHILDES (20M) | 77.77% | MOT |

### A.3.3 COMPUTE RESOURCES AND INFRASTRUCTURE FOR LANGUAGE MODEL TRAINING

Our language model experiments were run on a cloud provider VM instance consisting of four A100s (80GB VRAM each).

### A.4 VISION BENCHMARK DETAILS

### A.4.1 VIDEO PREPROCESSING

**BabyView** We sample BV-Home at 5 FPS at a resolution of 720x360 for the initial 224 global crop training of DINO, and at 720x1280 for the 518 high resolution final stage of training. This results in a total of 8M frames.

To create datasets of different sizes (1%, 5%, etc.) we randomly select complete clips and append them to a continuously increasing list which we save at different size increments. This ensures that every smaller set of data is a strict subset of the larger set (e.g., the clips in the 1% set are all contained in 5% set etc.). After getting these lists of clips, we extract frames with the same procedure.

---

[3] https://github.com/babylm/evaluation-pipeline-2023

Because the dataset is at a 9:16 widescreen aspect ratio, significantly different from the mostly 4:3 ImageNet image aspect ratio for which the DINO random cropping strategy was developed, we take random crop with aspect ratio in the 4:3 to 3:4 range with the biggest possible size, before performing the DINO cropping and augmentation. Empirically this results in a 1% improvement in ImageNet classification accuracy.

**SAYCam**   We sample SAYCam at 5 FPS in the native resolution of 480x640. This results in a total of 8.5M frames.

**Ego4D**   We take the complete Ego4D dataset without additional post-processing and sample frames at 1 FPS using ffmpeg at 1/2 of the original resolution. The smallest side of the images we extract ranges from 360 to 960 pixels—sufficient resolution for training (the variance in resolution exists in the original dataset due to the use of different recording devices). We reduce the original resolution to reduce the footprint of the dataset on disk and to lower the computational cost of data loading. This results in a total of 15M frames. We apply the same 3:4 aspect ratio augmentation that we did for BabyView.

### A.4.2   TRAINING

**DINOv2**   To train DINOv2 we use the official code repository.[4]  We try to perform minimal modifications of the existing pipeline. We train a ViT-B/14 with a batch size of 1024 with the default ImageNet-1K training config for the default 125K parameter updates. This initial training is done with a global crop of 224x224. All other hyperparameters are kept the same. We experimented with doubling the amount of parameter updates but did not see improvements. Following the DINOv2 paper, we train for an additional 10K parameter updates with a global crop of size 518x518.

**MoCov3**   To train MoCov3 we use the official code repository.[5]  We train a ViT-B/16 with a batch size of 512 with the default ImageNet-1K training configurations for up to 725K parameter updates. Similar to DINOv2, the training is done with an initial global crop of 224x224.

### A.4.3   DOWNSTREAM TASKS

**ImageNet Category Recognition**   We use the code from the official DINOv2 repository for kNN classification or for training a linear classifier. Our evaluation procedure, therefore, directly follows the procedure used in DINOv2.

**NYUv2 Depth Estimation**   Following the descriptions in the DINOv2 paper, we use the Monocular Depth Toolbox (Li, 2022). The code interfacing DINOv2 with this package is not released, but the trained depth estimation models and configs are released. After writing the interface code, we verify that the evaluation is correct by training a DPT-based depth estimator using this codebase on top of an off-of-the shelf official DINOv2 checkpoint which matched the performance from the paper.

**COCOStuff Semantic Segmentation**   We interfaced the official DINOv2 code with the mmsegmentation package (Contributors, 2020b). Similarly, the interface code is not released but the models and configs are available. To verify correctness, we trained a linear probe on top of an off-the-shelf official DINOv2 checkpoint and matched the performance from the paper on PASCAL VOC. We used the same config to train a linear probe on COCOStuff as was released for PASCAL VOC. We did not find improvements by training for longer. Future work may investigate training more complex architectures, which was prohibitive for this work due to the time and compute constraints required.

### A.4.4   COMPUTE RESOURCES

The DINOv2 vision models in this paper can be trained on a single 8x NVIDIA A40 GPU node. While no multi-node training is required, one full training run of DINOv2 takes about 3 days on 8x A40 GPUs. This translates to about 550 GPU hours per experiment, making it difficult to perform multiple runs to obtain error bars.

---

[4]`https://github.com/facebookresearch/dinov2`
[5]`https://github.com/facebookresearch/moco-v3`

### A.5 DATA ACCESSIBILITY

No data is available for review due to the parental embargo policy. All data will be hosted on https://nyu.databrary.org/ in November 2024 after the parental embargo period has lapsed. Researchers must be affiliated with a PI at a research-institution, who must request access to the project.

All compressed videos and their associated meta-data will be named according to a standardized format that encodes the subject id and the date at which the recordings were made. A .csv spreadsheet will provide detailed, anonymized information about each individual participant. Separate language outcome data (in standard CDI format) will be provided and linked to the individual subject IDs.

### A.6 LICENSING

The code and behavioral data published with the benchmark will be licensed under CC BY-NC 4.0. The video dataset is licensed under the terms laid out in the Databrary Access Agreement, see https://databrary.org/about/agreement/agreement.html.

License for Annotation models: YOLOXPose is licensed under the GPL-3.0 license. MMPose, RTMO, SimCC, ViTPose, mmsegmentation, DINOv2, Monocular Depth Toolbox, and LabelStudio are licensed under the Apache-2.0 license. GPT-2 is licensed under the modified MIT License. RTMPose is licensed under the MIT license. All are permissive for this paper release.

We the authors bear all responsibility in case we have violated any rights by the publication of these data and code in these venues.

### A.7 CODE AVAILABILITY

Anonymized, relevant model training code can be found at https://tinyurl.com/osf-babyview-codebase.

