# OpenReview forum: "The BabyView dataset: High-resolution egocentric videos of infants’ and young children’s everyday experiences"
_ICLR.cc/2025/Conference — Submitted to ICLR 2025_

### Official Review · Reviewer_FARf · 2024-10-31

**Soundness:** 2
**Presentation:** 3
**Contribution:** 4
**Rating:** 5
**Confidence:** 4

**Summary:**

In this paper, 2 new datasets are proposed consisting of recordings from children growing up between 6 months and 3 years representing a huge effort from collection. This specialised egocentric viewpoint represents a rare collection. The babyview dataset includes over 430 hours from multiple children and a separate 47 hour dataset which consists of one participant is also released (named Ego-SingleChild). Experiments showcase the interest in the data collected for self supervised models, and evaluating on language understanding tasks as well as vision tasks such as human pose estimation and object recognition.

**Strengths:**

* The data collected is interesting, will be released to the research community, and represent new video data that can be used for both computer vision and other disciplines (such as psychology).
* The effort for the data collection is huge representing long-term, real world data collection totalling over 400 hours of recorded video.
* The data could be of great use for further self-supervised research and model/algorithmic improvements.

**Weaknesses:**

# Weaknesses

* Table 1: EPIC-Kitchens and part of Ego4D should have Accelerometer data from the recordings depending on the devices that are used from the papers of both datasets. Additionally, what does the long category mean in this context? What's the threshold for a dataset to be considered long?
* Line 129 describes the work as the first release of the dataset. This phrase makes me question as a reviewer whether the dataset can be considered finished, is the work complete enough to make use of for the proposed tasks or should the community wait for more data? This statement feels
* Line 152, what does 0;5-3;1 years mean in the context of the ages?
* Table 3 AP and AR are not defined (this is only defined in the context of the appendix however there is a lot of space within the main paper for more definitions such as these.
* The experiments showcase interesting results, but they feel a bit unfocused as a section, it's not clear what the main tasks are for this dataset, beyond what is discussed within the paper, but then in the discussion section speech recognition and pose recognition perform well on the data from the dataset.
* Line 425 I'm not sure I agree with the statement of providing improved data, and not sure where this is coming from for the results.
* Line 733: Some of the demographics seem very skewed towards certain demographics. This is not a failing of the dataset as it represents initial work and findings, but to have some discussion/experiments regarding this lack of variance would be good to see. Of particular note is the family income.
* Line 802: The speaker labels imply that only mothers were captured during the dataset, is this true?
* It would be interesting to see whether there is a large domain shift between the babyview dataset and the Ego-SingleChild, as well as the other egocentric datasets that have been compared to in the paper.

# Other Comments
Table 1 has motion misaligned with the rest of the table.

**Questions:**

1. I would like to see some clarification regarding what the authors envisage for the future of the dataset and any specific tasks that they have in mind. Maybe it is because of the current structure of the paper, but this wasn't clear to me over my multiple read-throughs. The experiments consist of a mix of potential tasks and exploratory experiments for the reader to get a sense of the dataset. This is only mentioned within the discussion, but it's hard to refer to which experimental results refer to which part of the discussion.
2. Why is the current version of the dataset being released now, if data is continued to be collected, why not wait until a larger dataset can be released? There are some experiments within the paper that showcase the amount of data required, i.e. Figures 2 and 3, imply that more data is needed, but then the discussion implies that more data wouldn't help.
3. Have experiments/more investigation gone into the potentially skewed demographics that have been collected? It would be good to see some discussion regarding these.
4. Were only mothers and children collected within the dataset? Was there a reason as to why fathers were not included?
5. Have any experiments/investigation regarding the difference in domains between the different cameras from the two datasets as well as other egocentric datasets such as Ego4D or EPIC-Kitchens?

**Details Of Ethics Concerns:**

None, the paper has a lot of information regarding the collection and ethical protocol during the collection.

---

> ### Author Response · Authors · 2024-11-24
>
> Thanks for engaging with the paper, and for giving us the opportunity to clarify why we see this dataset – both the first release and the ongoing collection – of this dataset as important. First, please see the response to Reviewer #F6HH, where we elaborate on the utility of this dataset for the ML community.
>
> We would also like to emphasize that the first release of our dataset is still the largest and the most diverse developmental egocentric dataset to date—in fact, for scientists seeking to model and understand human cognition, this is likely more important than the fact that these videos are higher-resolution videos than older datasets or contain accelerometer/gyroscope data. The SAYCam dataset (Sullivan et al., 2020), while unprecedented in size at the time, only captured everyday experiences from three children of psychologists (who were also the first children in their families), limiting the scope of the generalizability of the conclusions that can be drawn from these data (this is also true of the Ego-SingleChild subset). In contrast, the full BabyView dataset seeks to capture the child’s perspective across a much greater number and more diverse set of families across the U.S.
>
> With respect to the family demographics, we note that the BabyView videos include many families where both speech from mothers/fathers/grandparents and other care-takers occurs. The example in the SI just happened to highlight the speech of  a mother. Some of our families do have relatively higher incomes; recruiting low income families remains a major challenge in developmental research (Singh et al., 2023). As we build the BabyView dataset, we are actively engaged in recruiting participants from a wide variety of backgrounds. By collecting from a combination of convenience samples as well as targeted samples of families from under-studied  backgrounds, our goal is to maximize the number of hours of data while striving for as representative a sample as possible. In future releases of the dataset, these demographic variables will be made available within Databrary.
>
> Because of the slow pace of data collection for BabyView, our goal is to release the dataset in chunks as it becomes available in order to maximize scientific progress across all research groups, not only within our own. We schedule each release after we have had time to clean each portion of the dataset and ensure that parents have been given ample opportunity to retract any portion of their videos.  We currently are planning to collect data through September 2025, pending funding, and will release all of the data via Databrary after the six month embargo expires (Spring 2026).
>
> As per your last comment on the differences in domains, please see our response to Reviewer es8L, which emphasizes that we elected not to evaluate image-trained models on egocentric video benchmarks.
>
> In our revised manuscript, we will (1) emphasize these points in both the Introduction and the General Discussion, (2) provide an overview of why we have chosen the particular annotation and training tasks to characterize the dataset as a first pass before diving into details, (3) provide additional clarifying details throughout the methods/results, including details on the datasets in Table 1,AP/AR (which refer to average precision/recall), the ages of the child participants (0;5 represents 0 years, 5 months, and this is a standard shorthand in developmental psychology papers), and the procedure for diarization, and (4) clarify why we chose not to  evaluate image-trained models on egocentric video benchmarks.
>
> References:
>
> Gilmore, R. O., Adolph, K. E., & Millman, D. S. (2016, August). Curating identifiable data for sharing: The Databrary project. In 2016 New York Scientific Data Summit (NYSDS) (pp. 1-6). IEEE.
>
> Sullivan, J., Mei, M., Perfors, A., Wojcik, E., & Frank, M. C. (2021). SAYCam: A large, longitudinal audiovisual dataset recorded from the infant’s perspective. Open Mind, 5, 20-29.

---

> > ### Comment · Reviewer_FARf · 2024-11-25
> >
> > I thank the authors for their response and time taken to address my concerns. The clarifications regarding the dataset collection and release schedule proposed. However, I am still unsure as to the tasks that are proposed for the dataset, the results section is still confusing to read and doesn't have a clear message in my opinion. Additionally, the lack of image/video trained models on the dataset is concerning and has not been addressed so far.
> >
> > I agree with reviewer BXkg08's last comment that a new version of the paper can be uploaded for further clarifications. The promised updates to the revised manuscript are good to see, but haven't included enough detail for me to consider as answering my question/addressing the weaknesses. Either seeing an updated manuscript or more detail in a response here would be useful.

---

### Official Review · Reviewer_H2M4 · 2024-11-01

**Soundness:** 2
**Presentation:** 3
**Contribution:** 3
**Rating:** 5
**Confidence:** 4

**Summary:**

This paper presents the BabyView dataset, a novel dataset of egocentric videos from infants aged 6 months - 5 years captured over their developmental years. It includes 430 hours of high-resolution videos captured from a GoPro Hero Bones camera mounted on a safety helmet designed for infants (100 x 75 degrees FoV) and gyroscope/accelerometer data. Furthermore, there are annotations for evaluating speech transcriptions, speaker diarization and human pose estimation. The authors perform language learning and visual task learning experiments to assess the quality of the data when compared to other egocentric datasets and internet-style curated datasets. The findings suggest that current self-supervised approaches are more successful when trained on curated datasets than on egocentric datasets. Overall, this dataset presents a challenge for developing human-like AI systems that can learn efficiently with sensory information captured from the developmental stages of infants.

**Strengths:**

I like the motivation behind the paper and feel it has some key strengths.
* The dataset contains diverse egocentric videos captured through the developmental stages of infants. It will be a valuable resource to the community for developing biologically-motivated learning algorithms.
* The large-scale nature and diversity are useful for splicing the dataset along different axes (e.g., data from a particular age group, data for a single infant through several years of development, etc.). Moreover, the dataset is not split into train/val/test splits. Different communities can split the data as appropriate for their settings.
* The authors have promised to share the data while maintaining the privacy of participants. This is missing in prior developmental datasets that can have restrictions, barring large parts of the research community from using them openly.

**Weaknesses:**

IMO, the paper has three main weaknesses.

# 1. Objective measures of the dataset's value

* The authors claim that this is a great dataset, but the objective value of the dataset is not quantified properly.

* First, none of the existing learning algorithms benefit from BV-Home (for language in Figure 2 and vision in Table 4). All ego datasets fall short of curated datasets. But more importantly, BV-Home does not distinguish itself from Ego4D / SayCam / CHILDES in any way. All the analyses presented could have been equivalently performed on other datasets (e.g., SayCam), and similar conclusions would have been reached.

* L431 "our results suggest that current self-supervised ... are dependent on large, curated datasets .." -- this suggests that the failure of current methods to do well with BabyView is attributed to the method design and not the dataset design. While this is partly true, it remains to be seen whether more developmentally accurate learning algorithms (whatever they may be) would do well with the BabyView dataset. As acknowledged by the authors in L450 - 458, there are inherent selection biases and diversity restrictions associated with the dataset that could prevent developmentally accurate learning algorithms from working well.

* As the authors state in L445, developmentally appropriate benchmarks may be needed to quantify the value of the BabyView dataset. Since these do not exist yet, I feel it is up to the authors to demonstrate the utility of the proposed dataset. This is missing.

Overall, the benefit of BabyView dataset over existing datasets has not been demonstrated experimentally. If standard vision and text benchmarks are not appropriate, then the onus is on the authors to identify/design better benchmarks or alternative metrics to bring out the value of developmental ego video datasets.


# 2. Annotation quality and utility are not clear
* The results in Table 2 suggest that the WER (%) is low (which is good), but the diarization precision and recall are poor for "gold-standard annotations". The quality of the pose annotations is also not quantified.
* Moreover, the utility of these annotations is not clear. For example, the annotations in Ego4D were explicitly used to define benchmarks for the community to make progress on. The application in Ego4D was for head-worn AR/VR devices, which are a practical use case for the benchmarks. In BabyView, the annotations are stand-alone, and their utility is unclear and unquantified.

# 3. Tools to access the dataset
* As a quality-of-life measure for users who want to utilize this dataset, it is not clear how access to such a large-scale dataset will be facilitated and simplified. Will there be tooling developed to easily download and visualize the videos and annotations? Will there be tooling for users to download meaningful subsets of the dataset (e.g., data for infants from ages 1 - 2)?

**Questions:**

1. What is the objective measure of the dataset's value?
2. Is the annotation quality truly "gold-standard"? The vision community has a high bar for human annotation quality.
3. How do users navigate this large-scale dataset? Will appropriate tooling be developed around the dataset?

**Details Of Ethics Concerns:**

Since this dataset contains personally identifiable information (especially concerning infants), I feel an ethics review is warranted.

---

> ### Author Response · Authors · 2024-11-24
>
> Thanks for your comments and for engaging with the paper. First, please see our responses to Reviewer F6HH and Reviewer es8L, where we elaborate why we think this dataset is of value for researchers in the machine learning community and has scientific value more broadly as well.
>
> Your review highlights that our current models do not show a difference in results using data trained on older, lower-resolution datasets (like SAYCam) vs. our more diverse dataset (BabyView). We also found this surprising! This runs counter to the hypothesis that many researchers held (including us!) that we might see dramatically better performance with more data and with images from better, higher-resolution videos. The fact that this was not the case further puts the onus on algorithmic development for the field to make progress on building algorithms that can learn as humans do; any model that is supposed to learn the way humans do should perform better when it is provided with a more complete set of “training data” from the child’s perspective. In a revised paper, we will emphasize that the primary contribution of the paper is the dataset itself —which stands a challenge to the community—rather than any of our particular modeling results.
>
> We will also clarify why we believe our annotations are of particular value. The gold-standard annotations that we will provide are careful, hand-annotated, gold-standard data from ourselves looking at the data that can then be used to benchmark both speech transcription and pose annotation algorithms. To be clear, we will provide all model annotations as well as gold-standard annotations for the subset that was hand-annotated by humans. With respect to pose annotations, we will clarify that the highest performing models reach comparable performance with these egocentric frames vs. state-of-the-art benchmarks on allocentric images. With respect to the diarization results, we will also clarify that diarization was conducted with a state-of-the-art model (Lavechin et al., 2020), noting also that there are very few diarization models designed to handle child and child-directed speech, which are out-of-distribution for most speech processing models. Furthermore, the lower precision annotations were primarily for child speech, in which the model typically confused the key child and other children; this is perhaps unsurprising given that child utterances tend to be much shorter (and thus harder to identify).
>
> Finally, with respect to access to the dataset, we will be providing access via Databrary, a well-established database platform for video data which provides a user-friendly interface that allows for easy viewing and browsing of the dataset (e.g., videos are playable within the browser) and selection of individual videos by demographic and participant variables. We will be providing detailed metadata to accompany the dataset, including family demographics as well as parent-report vocabulary scores for a subset of the children for whom it’s available. The metadata as well as the public Databrary API will allow researchers to download and process particular portions of the dataset that meet their own research requirements or interests. Finally, in a revised paper, we will clarify the procedures that researchers should follow to access these videos and their associated metadata.
>
> References:
>
> Lavechin, M., Bousbib, R., Bredin, H., Dupoux, E., & Cristia, A. (2020). An open-source voice type classifier for child-centered daylong recordings. arXiv preprint arXiv:2005.12656.
>
> Gilmore, R. O., Adolph, K. E., & Millman, D. S. (2016, August). Curating identifiable data for sharing: The databrary project. In 2016 New York Scientific Data Summit (NYSDS) (pp. 1-6). IEEE.
>
> Lasky, E. Z. & Klopp, K. (1982). Parent–child interactions in normal and language-disordered children. Journal of Speech and Hearing Disorders, 47: 7–18.

---

> > ### Comment · Reviewer_H2M4 · 2024-12-03
> > **Official Comment by reviewer H2M4**
> >
> > I thank the authors for their responses and clarifications. My concerns have been partly addressed, but key questions still remain. I am inclined to retain my rating of 5.
> >
> > * Data access/quality of life concerns have been partly addressed; the authors have stated the features they intend to make available. However, as far as I can tell, the procedures to access the videos and metadata have not been updated in a revised paper.
> > * Thanks for clarifying the differences b/w gold-standard human and machine annotations. However, my broader concern remains. The lack of gold-standard human annotations on a large scale and the errors in machine annotations reduce the value of the annotations.
> > * Value of dataset concerns: I understand the authors' point of view that current algorithms may not be designed to work with developmental data. It is entirely possible that better algorithms may benefit from BV-Home. However, the value of the dataset amongst other egocentric datasets remains questionable. There is no evidence that BV-Home can serve as a better developmental dataset than existing egocentric video datasets (developmental or otherwise). This needs to be quantified objectively.
> >
> > Like other reviewers, I also encourage the authors to upload a revised version of the paper that addresses these concerns.

---

### Official Review · Reviewer_es8L · 2024-11-03

**Soundness:** 3
**Presentation:** 3
**Contribution:** 3
**Rating:** 8
**Confidence:** 4

**Summary:**

This paper presents the babyview dataset, a developmental dataset with egocentric videos taken from the child’s perspective. Alongside, language and human pose annotations are provided. Furthermore, the authors benchmark language and visual self-supervised learning on this dataset, providing insightful discussions on the results.

**Strengths:**

+ The BabyView dataset is a valuable addition to the limited pool of egocentric developmental datasets and is well-positioned to enable further analysis in this domain.

+ The benchmark results in Section 5 and the general discussion in Section 6 are particularly engaging. They highlight the challenges in visual self-supervised learning from egocentric videos in particular, which lag behind language representation learning.

+ It is good to have the data scaling experiments (Figure 2 & 3). They provide an estimation of the data amount needed for machines to achieve human-level performance, offering insights for future research.

**Weaknesses:**

The paper notes the greater difficulty of visual self-supervised learning on egocentric videos compared to language representation learning. While this observation is generally accurate, additional discussion could clarify the role of domain gap in downstream tasks. Namely, language evaluation is less impacted by domain shift, unlike visual tasks, where the evaluation tasks (object detection, depth estimation, semantic segmentation) are essentially from third-person viewpoints. This disparity impacts the performance of models trained on egocentric data. On the other hand, several egocentric tasks in EPIC-Kitchens and Ego4D, such as egocentric action recognition, could also serve as evaluation tasks. These may show different trends, as they do not face the ego-exo viewpoint gap. It would be helpful if the authors could expand on these distinctions and their implications.

**Questions:**

+ Including EgoExo4D (https://arxiv.org/abs/2311.18259) in related works and Table 1 for comparison could be helpful, as it’s a recently introduced foundational benchmark for egocentric research. Additionally, could the authors clarify what the "Long?" and "N" columns in Table 1 signify?

+ Section 4.2, are the pose annotations specifically for adults recorded in the videos? An example figure of pose annotations in the main paper could enhance clarity. Likewise, it would also be good to include a language annotation example in the main paper.

+ L356-358, Could the authors explain the rationale for extracting Ego4D at 1 FPS versus BV-Home and SAYCam at 5 FPS?

---

> ### Author Response · Authors · 2024-11-24
>
> Thanks for your comments and engaging with our paper. We generally agree with many of your comments but we want to highlight that we see the primary contributions of our paper to be the dataset. The BabyView dataset is the largest and most diverse dataset of egocentric videos from the child’s perspective. These data are costly and time-consuming to gather, but – as we highlight in our response to Reviewer F6HH above – they represent our best estimate of the true training data for natural intelligence (babies). Any model that is supposed to learn the way humans do must be able to succeed using human data!
>
> Your review highlights that there are many possible learning targets to evaluate based on egocentric video. In the vision domain, our paper focused on transfer to third-person object recognition because we know that babies are good at object recognition and this task has become an important and challenging transfer task in the literature (e.g., Zhuang et al., 2020; Vong et al., 2024; Orhan & Lake, 2024). Babies do not show any “domain gap” as far as we know – even though they are “trained” on continuous views of the world from an egocentric perspective, they have no trouble recognizing objects in pictures (e.g., Deloache et al., 1979)!
>
> Nevertheless, you are right that there are other possible video benchmarks that we could use that would require less domain shift, including several from EPIC-Kitchens and Ego4D. We have not yet trained video models on the BabyView data, so we elected not to evaluate on these benchmarks, but we view this as an important next step.
>
> In a revised paper, we will also take care to clarify some of the methodological points raised in this review. We will add EgoExo4D as an additional dataset to the table, and clarify that “long” refers to “longitudinal” (videos by the same person over time). In addition, we will add details on the pose annotations, which were manually completed by an author on a small portion of frames in order to evaluate the pose detection models mentioned in the text. We will show pose annotations on example frames from families who have provided public access to their videos; we are able to share this video which has the automated annotations overlaid on a sample video. In addition, we will clarify that we extracted Ego4D at 1 FPS in order to have a comparable amount of frames across training regimes between this dataset and the developmental datasets.
>
> We extracted Ego4D at 1 FPS to make it feasible to train on that data using our hardware I/O and capacity constraints. Semantic information in video is highly redundant at a small temporal scale. As a result, for image-based representation learning and tasks like category recognition or semantic segmentation, we expect that 1 FPS sampling will not result in meaningful loss of learning signal.
>
> References:
> DeLoache, J. S., Strauss, M. S., & Maynard, J. (1979). Picture perception in infancy. Infant behavior and development, 2, 77-89.
>
> Orhan, A. E., & Lake, B. M. (2024). Learning high-level visual representations from a child’s perspective without strong inductive biases. Nature Machine Intelligence, 6(3), 271-283.
>
> Vong, W. K., Wang, W., Orhan, A. E., & Lake, B. M. (2024). Grounded language acquisition through the eyes and ears of a single child. Science, 383(6682), 504-511.
>
> Zhuang, C., Yan, S., Nayebi, A., Schrimpf, M., Frank, M. C., DiCarlo, J. J., & Yamins, D. L. (2021). Unsupervised neural network models of the ventral visual stream. Proceedings of the National Academy of Sciences, 118(3), e2014196118.

---

> > ### Comment · Reviewer_es8L · 2024-11-26
> >
> > Thank the authors for providing the responses. After reviewing the rebuttal and the comments from other reviewers, I feel the paper could benefit from the following improvements: (1) providing a clear revised version that incorporates the promised changes, and (2) emphasizing the utility of the dataset, could be through demonstrating improved performance on downstream tasks (potentially from egocentric domains) or by defining and proposing benchmark tasks associated with this dataset.

---

### Official Review · Reviewer_F6HH · 2024-11-04

**Soundness:** 1
**Presentation:** 3
**Contribution:** 2
**Rating:** 3
**Confidence:** 4

**Summary:**

The authors propose a large-scale, high-resoultion dataset for visual learning for infants. The dataset contains large field-of-view, accelerometer data, and annotations including speech transcription, speaker diarization, and human pose estimation. Using this dataset, the authors investigated self-supervised models in both language and visual domain, and found inferior performance compared to models trained on curated datasets.

**Strengths:**

[+] The proposed dataset aims to provide data for a very interesting and important research question - How can biological vision learn so efficiently and effectively with limited amount of data (experience) and supervision.

[+] The authors provide detailed description of dataset preparation, including data capture, access, and annotations.

[+] It is interesting to see investigation into both the linguistic and vision representation learning jointly stemmed from a single dataset.

**Weaknesses:**

[-] My main concern is that the contribution and usefulness of the proposed dataset is not exactly clear. Although large-scale and high-resolution, the dataset does not contain any control or interaction motivated by the observer (infant).
- It is my belief that a large component of visual learning is from interacting with our environments and paying attention to things that surprise us [A].
- This feedback is absent in the datasets of this format. I am curious what the authors' position on the significance of the lack of observer-driven controls, and whether it is feasible for such a dataset to produce model that can match human performance.
- By simply observing the visual world from an infant perspective, I am leaning towards the results from the Kitten Carosel experiment (Held And Hein, 1963).
- This perspective is further confirmed by the reported inferior quantitative results, both in language and visual domain.
- Although this dataset looks promising, I could not find sufficient evidence in the current version of the manuscript to support its usefulness. Further investigations in how this dataset can be useful for certain tasks / insights would greatly improve the paper in my opinion.

[-] Regarding the investigation in self-supervised vision models, the authors considered DINOv2 and MoCov3, which both assume a diverse set of images as input. I don't believe that they would take the full advantage of the video data in this dataset. Lines of work including [B-D] would be more appropriate here.

[A] Renee Baillargeon, Elizabeth S. Spelke, and Stanley Wasserman. "Object permanence in five-month-old infants." _Cognition_ 20.3 (1985): 191-208.

[B] Clément Godard, et al. "Digging into self-supervised monocular depth estimation." Proceedings of the IEEE/CVF international conference on computer vision. 2019.

[C] João Carreira, et al. "Learning from One Continuous Video Stream." _Proceedings of the IEEE/CVF Conference on Computer Vision and Pattern Recognition_. 2024.

[D] Shashanka Venkataramanan, et al. "Is ImageNet worth 1 video? Learning strong image encoders from 1 long unlabelled video." _arXiv preprint arXiv:2310.08584_ (2023).

**Questions:**

Please refer to weaknesses.

---

> ### Author Response · Authors · 2024-11-24
>
> Thank you for engaging with our paper. We believe that our viewpoint on learning is similar to yours, namely that both human and artificial intelligences likely learn best through active engagement with the world. Indeed, the BabyView corpus consists of videos from the perspective of infants actively engaged in exploration and interaction with the world, making them a rich source of insights about how this process unfolds.
>
> Babies in our dataset are in their own homes, looking at the things that they are interested in, selecting what to look at (presumably via the mechanisms posited in reference [A]). But remember, we are studying human babies, so we cannot “control” them or know exactly why they take the actions they take! That said, our dataset is the first dataset with children to include data from inertial motion units (accelerometer and gyroscope) to shed light on the role of the child’s actions in changing their visual exposure.
>
> Video data about human babies’ experiences are a critical part of understanding the growth of human intelligence: these videos are as close as the field of developmental psychology has come to capturing the natural training data that humans receive (Smith et al., 2015). So, these data stand as a challenge to current machine learning algorithms. Human children learn robust and generalizable representations for vision and language with similar data to those we present (see e.g., Gopnik, 2022). The lack of strong results from standard algorithms in either the visual or language domain is evidence that we need to make new innovations (perhaps including some from the works on video learning cited in your review).
>
> In sum, the importance of our data is that, if we want to build artificial intelligences that learn like humans, they must be able to learn from the same data that humans learn from.
>
> References:
> Gopnik, A. (2022). Children, creativity, and the real key to intelligence. APS Observer, 35.
>
> Smith, L. B., Yu, C., Yoshida, H., & Fausey, C. M. (2015). Contributions of Head-Mounted Cameras to Studying the Visual Environments of Infants and Young Children. Journal of Cognition and Development, 16(3), 407–419. https://doi-org.stanford.idm.oclc.org/10.1080/15248372.2014.933430

---

### Official Review · Reviewer_BXkg · 2024-11-08

**Soundness:** 3
**Presentation:** 4
**Contribution:** 3
**Rating:** 6
**Confidence:** 4

**Summary:**

The paper presents a large-scale developmental video dataset (the Babyview dataset), recorded using wearable cameras worn by young children. Babyview comes with video and IMU data, as well as automatically generated speech transcription, speaker diarization, and human pose data. Leveraging this dataset, the authors conducted a series of benchmarks on visual and language representation learning, leading to some interesting results and findings.

**Strengths:**

The Babyview dataset presents one of the largest developmental egocentric video datasets capturing children’s visual experience. This dataset offers the opportunity to address key research questions in developmental physiology and machine learning, e.g., the gap in learning efficiency between machines and humans.

The dataset, which represents a considerable investment of effort, will be accessible to authorized research teams, serving as a valuable resource to facilitate future research.

While the annotations are automatically derived from data, a comparison to human annotations was considered to quantify the annotation quality.

The paper is fairly well written and easy to follow. Technical details are well described.

The benchmarks are interesting, though many of their results are not very surprising.

**Weaknesses:**

I am not entirely convinced by the recording setup. The camera has a large vertical field of view, but a rather narrow horizontal field of view that seems to be much smaller than human vision system. While it is true that we do not have high resolution peripheral vision, ignoring some major portion of the field of view seems problematic for capturing the children's visual experience.

The experiment design and the presentation of the results could use some work.
* For example, Figure 2 included a "logistic fit" to extrapolate the trend of models trained on developmental data. There is little justification or explanation about this fit. Similarly, Figure 3 considered a log-linear extrapolation of the vision results. It is not clear if this extrapolation is reasonable.
* Another example is Table 4. The performance gap between models pretrained on egocentric videos and those trained on Internet images may stem from the domain shift between the two types of data. One possible approach to address this is to evaluate models pretrained on developmental egocentric videos using other egocentric video datasets.

**Questions:**

It would be a good idea to include some details about the camera, e.g., the field of view of the camera, resolution of the video, to make the paper self-contained.

The authors are encouraged to further elaborate how the dataset can be used by the machine learning community at large.

---

> ### Author Response · Authors · 2024-11-24
>
> Thanks for your review and for your acknowledgement of the effort involved in collecting this dataset and its general scientific value.
>
> In our revision we will absolutely attempt to make clear to the ML community why this dataset is valuable. Please see our response to Reviewer F6HH below.
>
> In our revised paper, we will provide better justification for camera selection decision (which was present in the journal paper that first described the BabyView camera; Long et al., 2024) as well as additional details about the camera (e..g, the field of view, and resolution of the video, as suggested). In brief, developmental psychologists have used head mounted cameras with different fields of view in their research, recognizing that the majority of attention is focused on the central parts of the field of view because head movements are roughly synchronized with eye-movements (Yoshida & Smith, 2008). But several groups have argued that a large vertical field of view is critical for capturing 1) what children are doing with their hands, which tends to be in the lower periphery, and 2) caregivers’ faces for social signals, which tend to be in the upper periphery (e.g., Long et al., 2022; Franchak & Yu, 2022). Thus, the BabyView camera is designed to capture these specific peripheral experiences.
>
> The use of a logistic fit is motivated by the nature of the BLIMP benchmark, which is a two-alternative forced choice format. Thus, scores must range between .5 (chance) and 1 (perfect accuracy). A logistic (sigmoid) is the appropriate functional form for extrapolation to avoid predictions above and below the range of the measure, and has been used in related work on scaling laws (e.g., Ruan et al., 2024).
>
> Finally, regarding your comment about image vs. video models, we agree that the difference in view types between egocentric images vs. photographs may have an impact on model accuracy.  However, there are two relevant dimensions of variation. First, egocentric vs. non-egocentric; second, image vs. video. Given that we only trained models on images, we did not evaluate them using video evaluation methods, although we intend to train and evaluate video models on this dataset in the future. Further, ImageNet classification accuracy provides a broad standard for comparison between our trained models vs. other models, allowing for direct comparison to the current state of the art (e.g., Orhan & Lake, 2024).
>
> References:
> Franchak, J. M., & Yu, C. (2022). Beyond screen time: Using head-mounted eye tracking to study natural behavior. Advances in child development and behavior, 62, 61-91.
>
> Long, B., Goodin, S., Kachergis, G., Marchman, V. A., Radwan, S. F., Sparks, R. Z., Xiang, V., Zhuang, C., Hsu, O., Newman, B. and Yamins, D.L., & Frank, M. C. (2024). The BabyView camera: designing a new head-mounted camera to capture children’s early social and visual environments. Behavior Research Methods, 56(4), 3523-3534.
>
> Long, B. L., Sanchez, A., Kraus, A. M., Agrawal, K., & Frank, M. C. (2022). Automated detections reveal the social information in the changing infant view. Child Development, 93(1), 101-116.
>
> Orhan, A. E., & Lake, B. M. (2024). Learning high-level visual representations from a child’s perspective without strong inductive biases. Nature Machine Intelligence, 6(3), 271-283.
>
> Ruan, Y., Maddison, C. J., Hashimoto, T. (2024). Observational scaling laws and the predictability of language model performance. arXiv preprint arXiv:2405.10938.

---

### Public Comment · ~Satoshi_Tsutsui1 · 2024-11-22
**Would you consider publishing the data without Databrary?**

This dataset appears to be highly valuable, and I am very interested in using it. However, I am afraid I cannot access it due to the constraints imposed by Databrary. Would you consider making the data available through an alternative platform?

When I attempted to access another dataset via Databrary, I encountered the following response from my institution:

> As you may be aware from our ServiceNow page, [Institution Name] has established a Research Data Access Committee (RDAC) to review researchers’ requests for dataset access. Since you are applying as a staff member of [Institution Name], which requires institutional endorsement, you must complete internal documentation and secure your supervisor’s approval as outlined in the undertaking form. Furthermore, as previously mentioned, the agreement involves an unlimited liability clause and foreign governing law that cannot be negotiated. Therefore, it will be necessary to provide: 1) a compelling justification for this request, 2) an explanation of how the PI and school can mitigate the risks related to foreign governing law and unlimited liability, and 3) strong support from both the Chair and Dean for RDAC review.

This is because the requirements imposed by Databrary is too strict. Applicants are required to obtain approvals from their supervisor, department chair, and even the university dean. These demands are particularly challenging for researchers seeking to explore datasets like the baby’s egocentric dataset as part of non-funded, exploratory academic activities, such as course projects.

Anyway, I hope the authors will make the data more publicly available after acceptance. It would be ideal if the data could be accessed directly by the requester and the authors, with a data protection agreement signed only by the involved parties, without requiring approvals from institutional authorities such as the dean.

---

> ### Author Response · Authors · 2024-11-24
>
> We appreciate that Databrary access is challenging, however, this is a necessary layer of protection for the privacy of the families involved. These families spend a considerable amount of time and energy recording private moments in their everyday lives, and we have an ethical and legal obligation to prioritize our participants' privacy and well-being in this research endeavor. To that end, Databrary is the state-of-the-art repository for sharing video data that contain identifiable video and audio information. The developers of Databrary have created a platform that ensures that the correct procedures are in place so that the confidentiality of the families are protected. As such, it is our best and only resource for sharing identifiable video data via our Institutional Review Board agreement.

---

> ### Public Comment · ~Satoshi_Tsutsui1 · 2024-11-28
>
> Thank you for your response. While I do not agree that Databrary is the ideal platform for sharing the dataset and have had negative experiences with them in the past, I will not argue this point here, as it is not relevant to this paper. Moreover, I fully understand the importance of safeguarding participants' privacy and agree that protecting the families’ confidentiality is a top priority. In light of this, would it be possible for you to release an anonymized version of the dataset (e.g., by blurring faces, removing other privacy-sensitive information, and muting any personally identifiable audio) without using Databrary?

---

### Public Comment · ~Yanlai_Yang1 · 2025-05-08
**Is there a Databrary Link for the dataset?**

I wonder whether the dataset has been released on Databrary. If so, is there a direct link where authorized researchers can access it (e.g. https://nyu.databrary.org/volume/564 for the SAYCam dataset)? If not, are there any plans in terms of when to release the data?

---

> ### Public Comment · ~Bria_Lorelle_Long1 · 2025-05-09
> **Release planned for this summer**
>
> Thanks for your interest in the BabyView dataset. Unfortunately, two issues prevent us from posting the dataset. First, our platform (Databrary.org) is currently upgrading their upload functionality and so we cannot upload at this time. Second, because our manuscript was rejected from ICLR we have prepared a larger release of the dataset for another conference and will be releasing these data mid-summer pending availability of upload functions.

---

### Meta-Review · Area_Chair_jxRk · 2024-12-20

**Metareview:**

This paper introduces a new dataset of egocentric videos (433 hours from 28 families, with children from 6 months to 5 years old), with accelerometer data, and automated speech transcription as well as pose estimation. The goal of the dataset is to see how ML models learn on the same "training data" as human children. The authors perform some experiments on the dataset, including pre-training GPT-2 on the language part, and pre-training ViT on the vision data. Results show that these models do not perform well in the data regime of this dataset. While reviewers agree that this is an interesting dataset studying an interesting question, there remains concerns on the experimental design and value of the dataset for the community in its current form. For example, current experiments do not show any benefit to using BV-Home on downstream tasks, or additional insights to extract using this data (besides data scaling experiments, but it is not clear that those need to be on this dataset specifically). One potential experiment could be to test the effect of training on images vs. videos (a point also brought up by another reviewer), or adding accelerometer data which was not leveraged by current experiments. Additional concerns are raised on whether tackling BV-Home really means that we have achieved ML models that learn the same as human children, given the additional interactions of children with the world not captured in this dataset. I think this could be a valuable dataset with these concerns addressed, and encourage the authors to take into account suggestions from reviewers.

**Additional Comments On Reviewer Discussion:**

First, all reviewers appreciate the introduction of a new dataset and the effort it takes to build these. The points raised by reviewers include concerns about the value of the dataset itself for the community and clarifications & suggestions for experiments mentioned above. While the authors did respond to the rebuttal, it seems many of the suggestions from reviewers have not been added to the paper revision at this point. At the end of the rebuttal, most reviewers still had remaining concerns and questions about this work.

---

### Decision · Program_Chairs · 2025-01-22

Reject